# Contrast-Enhanced Imaging Features and Clinicopathological Investigation of Steatohepatitic Hepatocellular Carcinoma

**DOI:** 10.3390/diagnostics13071337

**Published:** 2023-04-03

**Authors:** Kailing Chen, Yadan Xu, Yi Dong, Hong Han, Feng Mao, Hantao Wang, Xuhao Song, Rongkui Luo, Wen-Ping Wang

**Affiliations:** 1Department of Ultrasound, Zhongshan Hospital, Fudan University, Shanghai 200032, China; 2Department of Ultrasound, Xinhua Hospital, Shanghai Jiao Tong Universitity School of Medicine, Shanghai 200092, China; 3Department of Pathology, Zhongshan Hospital, Fudan University, Shanghai 200032, China; 4Department of Radiology, Zhongshan Hospital, Fudan University, Shanghai 200032, China

**Keywords:** steatohepatitic hepatocellular carcinoma, contrast-enhanced ultrasound, contrast-enhanced magnetic resonance imaging, follow-up

## Abstract

Steatohepatitic hepatocellular carcinoma (SH-HCC) is a distinctive histologic variant of HCC for the presence of steatohepatitis. This study intended to evaluate the contrast-enhanced imaging features and clinicopathological characteristics of 26 SH-HCCs in comparison with 26 age-and-sex-matched non-SH-HCCs. The frequency of obesity (34.6%, *p* = 0.048) and type 2 diabetes mellitus (23.1%, *p* = 0.042) were significantly higher in SH-HCC patients. As seen via B-mode ultrasound (BMUS), SH-HCCs were predominantly hyperechoic (65.4%, *p* = 0.002) lesions, while non-SH-HCCs were mainly hypo-echoic. As seen via contrast-enhanced ultrasound (CEUS), 96.2% of SH-HCCs exhibited hyperenhancement in the arterial phase. During the portal venous and late phase, 88.5% of SH-HCCs showed late and mild washout. Consequently, most SH-HCCs and all non-SH-HCCs were categorized as LR-4 or LR-5. As seen via magnetic resonance imaging (MRI), a signal drop in the T1WI opposed-phase was observed in 84.6% of SH-HCCs (*p* = 0.000). Notably, diffuse fat in mass was detected in 57.7% (15/26) SH-HCCs (*p* < 0.001). As seen via contrast-enhanced MRI (CEMRI), most of the SH-HCCs and non-SH-HCCs exhibited heterogeneous hyperenhancement in the arterial phase (80.8% versus 69.2%, *p* = 0.337). During the delayed phase, 76.9% SH-HCCs and 88.5% non-SH-HCCs exhibited hypo-enhancement. Histopathologically, the rate of microvascular invasion (MVI) was significantly lower in SH-HCCs than non-SH-HCCs (42.3% versus 73.1%, *p* = 0.025). The frequency of hepatic steatosis >5% in non-tumoral liver parenchyma of SH-HCCs was significantly higher than in non-SH-HCCs (88.5% versus 26.9%, *p* = 0.000). Additionally, the fibrotic stages of S0, S1 and S2 in SH-HCCs were significantly higher than in non-SH-HCCs (*p* = 0.044). During follow-up, although the PFS of SH-HCC patients was significantly longer than non-SH-HCC patients (*p* = 0.046), for the overall survival rate of SH-HCC and non-SH-HCC patients there was no significant difference (*p* = 0.162). In conclusion, the frequency of metabolism-related diseases in SH-HCC patients was significantly higher than in non-SH-HCC patients. The imaging features of SH-HCCs combined the fatty change and typical enhancement performance of standard HCC as seen via CEUS/CEMRI.

## 1. Introduction

Hepatocellular carcinoma (HCC) is the sixth most common malignant tumor with 906,000 newly diagnosed cases globally in 2020, which also represents the third leading cause of cancer-related deaths [1]. The incidence of HCC varies from ethnicity to region, with most patients burdened with chronic infections of hepatitis B virus (HBV), hepatitis C virus (HCV), metabolic-dysfunction-associated fatty liver disease (MAFLD), or alcoholic steatohepatitis [2]. MAFLD, which used to be named nonalcoholic fatty liver disease (NAFLD), is characterized by hepatocyte steatosis, steatohepatitis, Mallory’s body and fibrosis histologically [3]. Currently, MAFLD has already afflicted 30% of the population all over the world, even though MAFLD in only 5% of patients will evolve into MAFLD-related fibrosis or HCC [4]. Although chronic HBV infection is still a leading cause of HCC in China, MAFLD-related primary liver cancer has been a rising epidemic, leading to substantial morbidity and mortality [5].

According to the 2019 World Health Organization (WHO) digestive tumor pathological classifications [6], HCC is a genetically heterogeneous entity that can be subdivided into numerous specific variants such as clear cell, macrotrabecular-massive, steatohepatitic or fibrolamellar carcinoma; e.g., Steatohepatitic hepatocellular carcinoma (SH-HCC) is a distinctive histologic variant for the presence of steatohepatitis, including the ballooning of malignant hepatocytes, inflammation, Mallory-Denk bodies and pericellular fibrosis [7]. It has been reported to be tightly associated with HCV infection, type 2 diabetes mellitus (T2DM), obesity and underlying MAFLD [7,8]. Nonetheless, Yeh et al. [9] identified partial SH-HCC patients whose absence of a fatty liver may be driven by tumor-specific genetic alterations. They commented that SH-HCC can exist independently of a steatohepatitic background.

Although few studies regarding the radiological features of SH-HCC have been published, the contrast-enhanced ultrasound (CEUS) characteristics have never been mentioned previously [10,11,12,13,14]. CEUS is a non-invasive, radiation-free and cost-effective method for detecting focal hepatic lesions (FLLs), which has been recommended as a first-line method for the characterization of FLLs and for postoperative follow-up according to many guidelines [15,16].

Contrast-enhanced magnetic resonance imaging (CEMRI) permits the definitive diagnosis of HCC in high risk patients without pathologic confirmation [17]. Moreover, MRI is of important diagnostic accuracy for liver fat content [18]. Therefore, this study intended to explore the CEUS and CEMRI features as well as the clinicopathological characteristics of SH-HCCs in comparison with non-SH-HCCs.

## 2. Patients and Methods

### 2.1. Patients

This work was approved by the Ethics Committee of Zhongshan Hospital, Fudan University (approval no. B2021-051), and written informed consent was waived.

Twenty-six pathologically confirmed SH-HCC patients (23 males and 3 females, age range: 34–78 years old) who underwent contrast-enhanced imaging examinations before surgery from January 2013 to October 2021 were consecutively enrolled in the present study.

Inclusion criteria were as follows: (1) acquisition of CEUS and CEMRI before surgery within one month; (2) SH-HCC confirmed by the WHO diagnostic criteria [6]; (3) no local or systematic anti-tumor therapies before imaging examination.

Exclusion criteria included the following: (1) the target lesion was invisible when using a B-mode ultrasound (BMUS); (2) poor quality of CEUS or CEMRI data; (3) anti-tumor therapies before image acquisition.

A total of 847 non-SH-HCC patients who met the inclusion and exclusion criteria from the database of our hospital between January 2016 and September 2021 were collected as the matched group. To minimize the selection bias, a propensity score matching (PSM) study was accomplished by equating the two groups based on age and sex with a nearest neighbor 1:1 matching scheme. The caliper size of 0.2 was used as well. Finally, the matched cohort of 26 non-SH-HCC patients was utilized in this study.

### 2.2. Clinical Data

Preoperative clinical data including the patients’ age at surgery, sex, body mass index (BMI), history of hypertension, hyperlipidemia, type 2 diabetes mellitus (T2DM), hepatitis B virus (HBV) infection and hepatitis C virus (HCV) infection, serum alanine aminotransferase (ALT), aspartate aminotransferase (AST), total bilirubin (T-Bil), alpha-fetoprotein (AFP), carbohydrate antigen 199 (CA199) and carcinoembryonic antigen (CEA) levels were recorded.

### 2.3. CEUS Image Acquisition

CEUS examinations were performed by three experienced sonographers with over eight years’ experience of liver CEUS. All examinations were performed using LOGIQ E9 GE (equipped C1-5 convex array transducer), Mindray Reason 8 (equipped CA1-7A convex array transducer) and PHILIPS EPIQ (equipped with C5-1 convex probe). The BMUS scan of the whole liver was firstly performed to locate the suspected hepatic lesion. Then, a dose of 1.5–2.0 mL SonoVue^®^ was injected as the contrast agent via the cubital vein and was followed by a 5 mL saline flush. The CEUS was performed according to the World Federation for Ultrasound in Medicine and Biology (WFUMB) guidelines [19]. A clip that displayed the enhancement process of the target lesion was recorded continually for 2 min. All images were recorded and exported in DICOM format.

### 2.4. CEMRI Image Acquisition

MRI scans were performed using the 3.0 Tesla MR scanner (Siemens, Erlangen, Germany) with body-phased array coils. The liver scanning protocols consisted of T1-weighted (T1WI) in-phase and opposed-phase gradient echo sequences, T2-weighted (T2WI) fat-suppressed fast-spin echo sequence and diffusion-weighted imaging (DWI) with single-shot spin-echo echo planar sequence (b = 0, 50 and 500 sec/mm^2^) [20]. The gadopentetate dimeglumine (Bayer HealthCare, Berlin, Germany) was used as contrast agent. The contrast agent was injected at a dose of 0.1 mmol/kg at a rate of 1 mL/sec using a power injector, followed by a 20 mL saline flush. The arterial phase acquisitions were triggered automatically when the contrast agent reached the ascending aorta, and dynamic T1WI MRI at the portal venous phase (60 s) and late phase (180 s) was performed, respectively.

### 2.5. CEUS Image Analysis

The CEUS images were reviewed independently by two independent radiologists (with more than 10 years of abdominal CEUS) who were blinded to clinicopathologic data and CEMRI results. When disagreement occurred, consensus was reached after discussion. At first, the reviewers evaluated the following BMUS features: number of lesions, maximum diameter of the target lesion (for multiple lesions, the largest one was analyzed as the target lesion), echogenicity (hypo-, iso-, hyper- or mix-echoic), homogeneity (homogeneous or heterogeneous), shape (regular or lobulated), margin (well- or ill-defined) and presence of color Doppler flow signals. Then, the CEUS features were interpreted based on the Contrast-Enhanced UltraSound Liver Imaging Reporting and Data System version 2017 (CEUS LI-RADS v2017) [21]: the size of the lesion, the enhancement intensity (hypo-, iso- or hyper-enhanced) and its pattern (homogeneous, heterogeneous, rim or peripheral nodular) during the arterial phase (10–45 s) and the presence, time (<60 s, ≥60 s) and degree (mild, marked) of washout during the portal venous (30–120 s) and late phase (after 120 s) of CEUS.

### 2.6. CEMRI Image Analysis

MR images taken on a PACS workstation (GE Medical Systems Integrated Imaging Solutions, Waukesha, WI, USA) were analyzed by two radiologists (with 5 and 8 years of experience in liver imaging) in an independent manner. The reviewers were aware that there were lesions in the liver but they were blinded to the patients’ clinical histories. For multiple tumors, the largest one was evaluated. In disputatious cases, the consensus review was made for final decisions when disagreement occurred.

Via unenhanced MRI, the morphological features were as follows: the maximum diameter, number (single or multiple) and signals (hypointense, isointense or hyperintense) on TIWI, T2WI and DWI, signal homogeneity (homogeneous or heterogeneous), margin (well- or ill-defined) and shape (regular or irregular). Fat in mass was intuitively divided into diffuse (fat composition >50% of the lesion area) or focal (fat component <50% of the lesion area). Hemorrhage or necrosis in mass were also evaluated.

The CEMRI features on the basis of the LI-RADS v2018 were evaluated as follows [17]: enhancement intensity (hyper-, iso- or hypo-intensity) and its enhancement pattern (non-rim or rim) in the arterial phase, enhancement intensity (hyper-, iso- or hypo-enhancement) in portal venous and delayed phase and tumor capsule enhancement (complete, incomplete or no).

### 2.7. Histopathological Examination

For each case, the hematoxylin-eosin and immunohistochemical staining slides were reviewed by two experienced pathologists (with over 15 years of experience). The diagnosis of SH-HCC was defined by the 2019 WHO criteria [6]. The histological grade of HCC was classified based on the modified Edmondson–Steiner classification [22]. The MVI status was defined as M0, M1 or M2 according to the Practice and Guidelines of the Chinese Society of Pathology [23]. The liver fibrosis stage and necroinflammatory activity was graded using the METAVIR system [24].

### 2.8. Follow-Up

All patients were followed up with ultrasound or MRI regularly after surgery. The cutoff follow-up date was 31 October 2022. If patients’ postoperative examinations were conducted at a local hospital, they were consistently kept in touch with through mobile phone. Progression-free survival (PFS) was determined between hepatectomy and either the first recurrence of HCC or the date of the final follow-up. Meanwhile, overall survival (OS) was calculated between the operation time and the final postoperative follow-up or death.

### 2.9. Statistical Analysis

SPSS version 26.0 (IBM company, Armonk, NY, USA) software was used for statistical analyses. Descriptive statistics (median, interquartile range) were compared using Student’s t-test or the Mann–Whitney test. The χ2 test or Fisher’s exact test were used to compare categorical variables (frequency, percentage). The overall survival of SH-HCC and non-SH-HCC patients was analyzed using the Kaplan–Meier method and then compared using the log-rank test. All statistical tests were two-tailed and a *p* value < 0.05 was considered to be of statistical significance.

## 3. Results

### 3.1. Baseline Characteristics

The baseline characteristics of 26 SH-HCC patients and 26 non-SH-HCC patients matched by gender and age using PSM are listed in Table 1. There was a male predominance (84.6%) in SH-HCC patients with a median age of 65.5 years old. The underlying HBV and HCV infections were of no significant difference between the two groups (*p* > 0.05). A significantly higher frequency of obesity (34.6%, *p* = 0.048) and T2DM (23.1%, *p* = 0.042) were observed in the SH-HCC group, whereas the prevalence difference for hypertension and hyperlipidemia was not significantly different (*p* > 0.05).

As for the laboratory test, although the serum AFP level in SH-HCC patients was significantly higher than for non-SH-HCC patients (*p* = 0.047), there were no significant differences in CA199, CEA, ALT, AST or T-Bil seen between the two groups (*p* > 0.05).

### 3.2. BMUS and CEUS Features

SH-HCCs were solitary in 92.3% (24/26) patients and multifocal in 7.7% (2/26) of patients. The median lesion size was 30.5 mm (range: 12–128 mm). As seen via BMUS, SH-HCCs were predominantly hyperechoic (65.4%, *p* = 0.002) lesions with a heterogeneous internal echo, regular shape and well-defined margin, whereas non-SH-HCCs mainly manifested as heterogeneous hypo-echoic nodules. Blood flow signals were detected in 80.8% (21/26) of SH-HCCs, and the median resistance index was 0.67.

As seen via CEUS, 96.2% (25/26) SH-HCCs and 100% (26/26) non-SH-HCCs exhibited hyperenhancement in the arterial phase. A total of 61.5% (16/26) SH-HCCs showed a heterogeneous enhancement pattern, 30.8% (8/26) were homogeneous and enhanced and another 7.7% (2/26) lesions exhibited rim-like hyperenhancement. During the portal venous phase, 57.7% (15/24) of SH-HCCs and 69.2% (18/24) of non-SH-HCCs were hypo-enhanced, respectively. Subsequently, 88.5% (23/26) of SH-HCCs and 96.2% (25/26) of non-SH-HCCs became hypo-enhanced in the late phase. Hence, late and mild washout (≥60 s) was the commonest characteristic in both the SH-HCC and non-SH-HCC group. As for the CEUS LI-RADS category, most SH-HCCs and all non-SH-HCCs were categorized as LR-4 or LR-5 (88.5% and 100.0%, respectively). The BMUS and CEUS features of the SH-HCCs and non-SH-HCCs are displayed in Table 2.

### 3.3. CEMRI Features

As shown in Table 3, SH-HCCs mainly showed hypo-intensity on T1WI and hyperintensity on T2WI, and there were no significant differences between the two groups (*p* > 0.05). However, there was a signal drop in the T1WI opposed-phase which was observed in 84.6% of SH-HCCs, which was much higher than for the non-SH-HCCs (*p* = 0.000). As seen via a contrast-enhanced scan, most SH-HCCs and non-SH-HCCs exhibited heterogeneous hyperenhancement in the arterial phase (80.8% versus 69.2%, *p* = 0.337). During the delayed phase, 76.9% (20/26) of SH-HCCs and 88.5% (23/26) of non-SH-HCCs exhibited hypo-enhancement (*p* = 0.264). As a result, SH-HCCs and non-SH-HCCs were mostly classified into LR-4 or LR-5. Notably, diffuse fat in mass was detected in 57.7% (15/26) SH-HCC lesions, which was significantly higher than for non-SH-HCCs (*p* < 0.001). Hemorrhage and necrosis in mass were of similar frequency in the two groups (*p* = 0.350, *p* = 0.703, respectively). The contrast-enhanced images of SH-HCC are shown in Figure 1, and images of non-SH-HCC are displayed in Figure 2.

### 3.4. Histopathological Findings

Most SH-HCCs were moderately differentiated (grade II/46.2%, grade III/53.8%, respectively). The rate of microvascular invasion (MVI) was significantly lower in SH-HCCs than non-SH-HCCs (42.3% versus 73.1%, *p* = 0.025). Meanwhile, no significant differences between Ki-67, vascular invasion, satellite lesions or lymph node metastasis were observed between the two groups (*p* > 0.05).

The frequency of hepatic stestosis >5% in non-tumoral liver parenchyma of SH-HCCs was significantly higher than non-SH-HCCs (88.5% versus 26.9%, *p* = 0.000). Additionally, the fibrotic stages of S0, S1 and S2 in SH-HCCs were significantly higher than non-SH-HCCs (76.9% versus 50.0%, *p* = 0.044), whereas no significant difference was seen in the necroinflammatory activity of SH-HCCs and non-SH-HCCs. The histopathological Findings of SH-HCCs and non-SH-HCCs were listed in Table 4.

### 3.5. Outcomes of SH-HCCs

During follow-up, the overall survival rate of SH-HCC and non-SH-HCC patients was 82.6% and 87.9%, respectively. According to the survival function, there were no significant differences in SH-HCC and non-SH-HCC patients (*p* = 0.162) (Figure 3). However, the PFS of SH-HCC patients was significantly longer than for non-SH-HCC patients (*p* = 0.046) (Figure 4).

## 4. Discussion

Due to a prevalence of lifestyle-related diseases such as obesity, metabolic syndrome and risk factors for MAFLD, the incidence of SH-HCC is increasing rapidly worldwide [25]. In our study, the SH-HCC patients carried a higher rate of obesity and T2DM in comparison with non-SH-HCC patients. As seen via BMUS, SH-HCCs mainly presented as hyperechoic lesions with a regular shape and well-defined margin. After the injection of SonVue, they were rapidly hyperenhanced in the arterial phase and late washout during the late phase. A signal drop in the T1WI opposed-phase was characteristic of SH-HCC, suggesting prominent intratumoral fat deposition. As seen via CEMRI, the enhancement performance of SH-HCCs was similar to non-SH-HCCs. Histopathologically, the rates of hepatic steatosis (>5%) in non-tumor liver parenchyma were significantly higher in SH-HCC patients than in non-SH-HCC patients. There was no significant difference in overall survival between SH-HCC and non-SH-HCC patients.

We observed the rate of HBV and HCV infection in SH-HCCs which was of no significant difference between the two groups. Conversely, the HCV infection in SH-HCC patients has been reported to be significantly different from non-SH-HCC patients by Yamaoka et al. [10]. The reason may be that HBV infection is much more common in Chinese people [26]. Furthermore, the tumor size of SH-HCCs has been reported to be larger than in non-SH-HCCs [10,13], while the diameter of tumors of SH-HCCs and non-SH-HCCs was of no significant difference in the present study. The probable reason may be that chronic hepatitis B patients account for over half of the SH-HCC/non-SH-HCC patients; routine imaging examinations and laboratory tests are performed to detect cancer as early as possible [26].

Intratumor steatosis is one of the marked features of SH-HCC, and diffuse and focal fat in mass was observed in 57.7% and 42.3% of lesions, respectively. Inui et al. [13] reported that diffuse and focal steatohepatitic features were seen in 60% (12/20) and 40% (8/20) of SH-HCCs, and the results were in line with ours. Except for SH-HCCs, diffuse or focal fatty change are considered to be the characteristics of early HCC according to the European Society for Medical Oncology (ESMO) guideline [27]. As early HCCs dedifferentiate towards progressed HCCs, the blood supply transfer from portal venous to arterial is supplied gradually, resulting in hypoxia-induced fatty metamorphosis inside hepatocytes [28]. The MRI findings also seem to back this up. Our present study detected focal fat deposition in only 19.2% of non-SH-HCCs. Additionally, Sano et al. [29] observed fatty change in 11% of progressed HCC lesions.

Compared with CEMRI, CEUS provides real-time visualization of dynamic enhancement and washout performance during the whole enhancement phase [30,31]. In the arterial phase, the SH-HCCs mainly appeared to be homogeneous or show heterogeneous hyperenhancement, which was also known as “APHE” [19,21]. During the late phase, mild and late (>60 s) washout were also observed in 23 SH-HCCs. Consequently, 88.5% of SH-HCCs were categorized as CEUS LR-4 or LR-5, which share similar characteristics of non-SH-HCCs. Hepatic nodules classified into LR-4 are highly suspicious for HCC, and the specificity of the LR-5 category for demonstrating HCC was as high as 89~94% [21,32]. Hence, SH-HCCs exhibit typical enhancement characteristics of standard HCC, as seen via CEUS. To the best of our knowledge, this is the first study regarding the CEUS features of SH-HCCs based on larger case series.

Furthermore, two SH-HCC lesions in our study showed rim-like enhancement in the early arterial phase and marked washout during the late phase, mimicking hepatic metastasis. On this occasion, a core needle biopsy is recommended when necessary. In addition, three SH-HCCs continued to show hyper- or iso-enhancement in the late phase, which were misdiagnosed as hepatic angiomyolipoma (HAML), hepatocellular adenoma (HCA) or focal nodular hyperplasia (FNH), respectively. Additionally, Bo et al. [33] also observed a case of SH-HCC that was rapidly hyperenhanced in the arterial phase but showed an absence of washout in the late phase. This is important for differential diagnosis because almost all benign hepatic focal lesions show continuous hyper- or iso-enhancement in the late phase [15,19]. In the multistep process of tumorigenesis, the portal vein supplies the early HCCs, which exhibit atypical mild hyperenhancement in the arterial phase without washout in the late phase [19,32,34]. In contrast, HCAs may exhibit hypo-enhancement in the late phases for lack of portal vein branches [35]. The washout of HCCs in non-cirrhotic liver is also supposed to be concerned with their differentiation grade: the moderate differentiated nodules (grade II/III) exhibit mild washout during the late phase, whereas the well-differentiated lesions (grade II/III) are continuously hyperenhanced during the late phase [15,36]. However, the SH-HCCs in the present study were all moderately differentiated (46.2% of grade II, 53.8% of grade III, respectively). A steatohepatitis-like change in FNH and HAML may also show overlapping features with SH-HCC; meanwhile, the non-cirrhotic liver background of SH-HCC complicates the diagnosis [37]. Under such circumstances, the elevated serum level of AFP may help with suggesting malignancy. If not, histological verification should be performed for accurate diagnosis before surgery.

In the present study, the MRI characteristics of SH-HCCs were also fully analyzed. The most notable result was that a signal drop was observed in 84.6% of SH-HCCs in the T1WI opposed-phase compared with the in-phase. In addition, a diffuse fat change in mass was seen in 57.7% of SH-HCCs. Our results were in line with Inui et al. [13]. As seen via contrast-enhanced scan, even though 88.5% of SH-HCCs showed hyperintensity in the arterial phase, the remaining 7.7% of these lesions exhibited hypo-intensity. Hypovascular enhancement is characteristic of the progressed HCCs but not for early, fatty degeneration or well-differentiated HCCs [29,38]. That is to say, the fact that they are fat-containing coincides with the presence of enhancement intensity during the arterial phase, and the SH-HCCs with a dominant fat component show less conspicuous enhancement [13,39]. During the portal venous and delayed phase, 76.9% of SH-HCCs started to show hypo-intensity. As a result, 80.7% of SH-HCCs were classified into LR-4 or LR-5. In a word, SH-HCCs presented the characteristics of fat degeneration as well as enhancement performance of standard HCCs.

Regarding the non-tumor hepatic parenchyma, steatosis was significantly different between SH-HCCs and non-SH-HCCs. This phenomenon has also been described in previous studies [10,25]. Importantly, we found that the liver fibrosis stage of SH-HCCs was lower than for non-SH-HCCs, and the results were similar to a multicenter study by Taniai et al. [11]. Microvascular invasion (MVI) refers to the presence of tumor cells in the peritumoral of surgical specimen via microscopic examination, which has been considered as a well-known prognostic parameter of early recurrence after curative resection [40]. We defined the MVI negative nodules in the SH-HCC group as being significantly higher than in the non-SH-HCC group. During follow-up, the PFS of SH-HCC patients was significantly longer than for non-SH-HCC patients, although the OS was of no significant difference between the groups. Generally, the prognosis of HCC remains dismal, with a 5-year survival rate of less than 20%, but the overall survival of those with SH-HCC remains to be further studied [6,28].

Several limitations should be stated. First, although our study enrolled participants during a long period of time, the SH-HCC patients were limited. Second, only surgically resected patients were evaluated, and selection bias may exist. Last, due to limited cases of SH-HCCs, a subgroup analysis of imaging features was not performed.

In conclusion, the frequency of metabolism-related diseases in SH-HCC patients was significantly higher than for non-SH-HCC patients. The imaging features of SH-HCC combined the fatty change and typical enhancement performance of non-SH-HCC, as seen via CEUS/CEMRI.

## Figures and Tables

**Figure 1 diagnostics-13-01337-f001:**
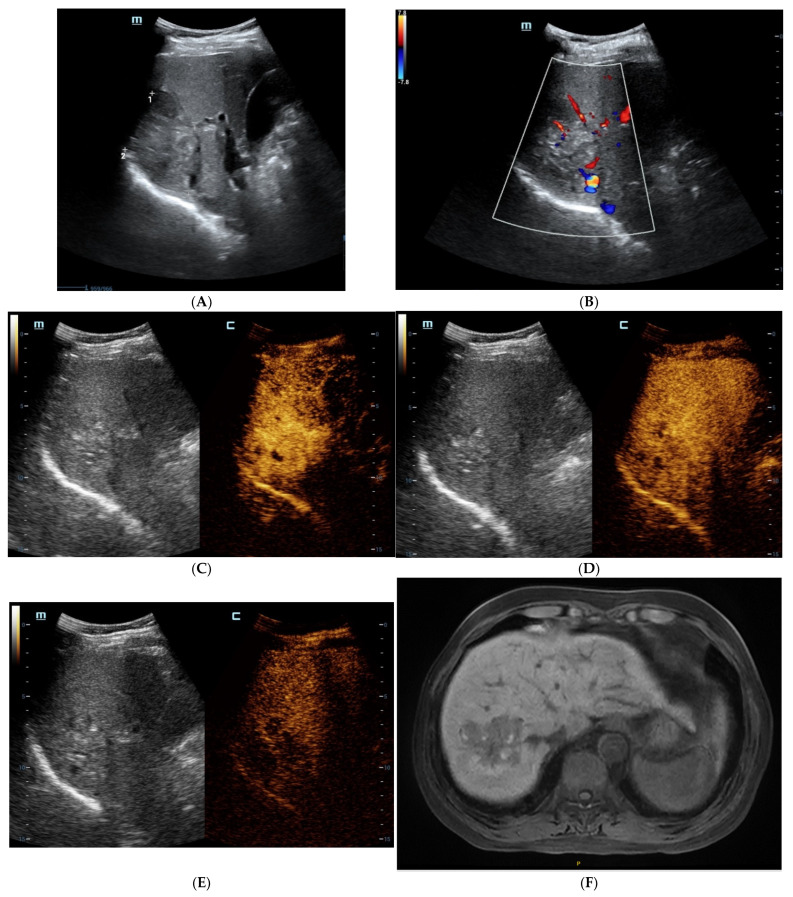
The contrast-enhanced ultrasound (CEUS) and contrast-enhanced magnetic resonance imaging (CEMRI) of steatohepatitic hepatocellular carcinoma (SH-HCC). A heterogeneous hyperechoic le-sion was detected in the right lobe of the liver (**A**). Short linear blood flow signals were seen in the peripheral of mass (**B**). As seen via CEUS, the hepatic nodule did not show rim, peripheral dis-continuous or heterogeneous hyperenhancement in the early arterial phase (**C**). During the portal venous phase, it was continuous iso-enhancement (**D**). It exhibited mild washout in the late phase (**E**). The hepatic lesion was heterogeneous hypointense on T1WI (**F**). A signal drop in the T1WI opposed-phase was observed (**G**). It was marked as hyperintense on unenhanced T2WI (**H**). After injection of contrast agent, the hepatic lesion showed non-rim hyperenhancement in the arterial phase (**I**). It became hypo-intensity during the late phase (**J**).

**Figure 2 diagnostics-13-01337-f002:**
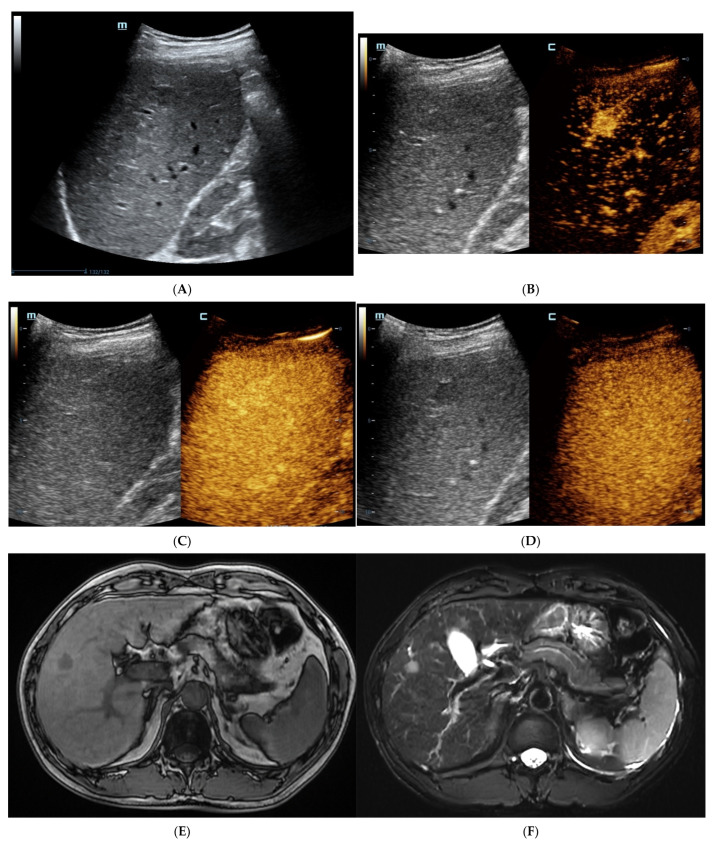
The contrast-enhanced images of non-steatohepatitic hepatocellular carcinoma (non-SH-HCC). A hypo-echoic lesion in the right lobe of the liver was seen as seen via BMUS (**A**). In the early arterial phase of CEUS, the hepatic nodule was homogeneously hyperenhanced (**B**). During, it presented continuous iso-enhancement during the portal venous and late phase (**C**,**D**). On T1WI, the hepatic nodule was homogeneous hypointense (**E**). On T2WI, it was hypointense (**F**). As seen via con-trast-enhanced scan, the lesion showed hyperintensity in the arterial phase (**G**). It exhibited slight hyperintensity in the portal venous phase (**H**). During the late phase, the hepatic lesion started to show iso-intensity (**I**).

**Figure 3 diagnostics-13-01337-f003:**
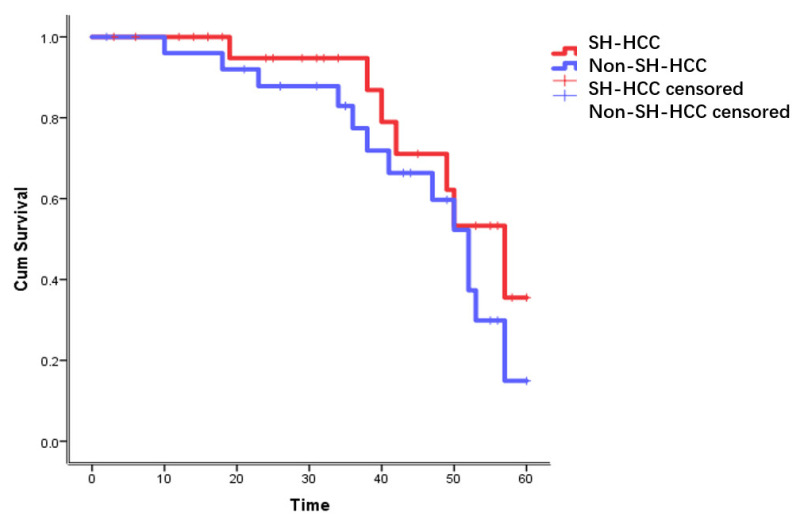
The survival curve of SH-HCC and non-SH-HCC patients.

**Figure 4 diagnostics-13-01337-f004:**
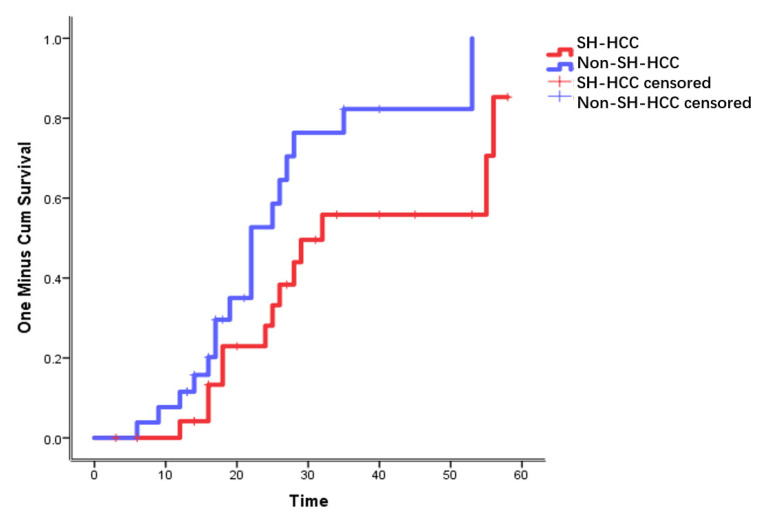
The recurrence curve of SH-HCC and non-SH-HCC patients.

**Table 1 diagnostics-13-01337-t001:** Baseline characteristics of SH-HCC and non-SH-HCC patients.

Variables	SH-HCC (*n* = 26)	Non-SH-HCC (*n* = 26)	*p*-Value
Age (years)	65.5(39, 74)	61.5(39, 72)	-
Male/Female, *n* (%)	22(84.6)/4(15.4)	22(84.6)/4(15.4)	-
Obesity, *n* (%)	9(34.6)	3(11.5)	0.048
T2DM, *n* (%)	6(23.1)	1(3.8)	0.042
Hypertension, *n* (%)	3(11.5)	5(19.2)	0.703
Hyperlipidemia, *n* (%)	5(19.2)	2(7.7)	0.419
Child–Pugh A/B, *n* (%)	25(96.2)/1(3.8)	24(92.3)/2(7.7)	1.000
HBsAg (+), *n* (%)	14(53.8)	21(80.8)	0.071
HCV-RNA (+), *n* (%)	2(7.7)	0	0.490
AFP (ng/mL)	155.0(37, 964)	283.5(28, 2319)	0.047
CA199 (U/mL)	9.0(3.5, 23.7)	13.4(0.6, 69.7)	0.355
CEA (ng/mL)	2.4(0.5, 34.3)	3.7(0.9, 23.5)	0.532
ALT (IU/L)	32.7(26.6, 187.3)	29.5(32.9, 232.8)	0.701
AST (IU/L)	37.4(23.9, 282.5)	35.6(30.8, 315.3)	0.085
T-Bil (mg/dL)	4.7(1.9, 76.8)	3.9(2.5, 85.6)	0.611

Note: Obesity: BMI > 25.

**Table 2 diagnostics-13-01337-t002:** Imaging features of the SH-HCCs and non-SH-HCCs as seen via BMUS and CEUS (%).

Features	SH-HCC (*n* = 26)	Non-SH-HCC (*n* = 26)	*p*-Value
Number of lesions (single/multiple)	24(92.3)/2(7.7)	26(100.0)/0	0.490
Diameter (mm)	30.5(18.5, 49.0)	42.5(22.5, 65.5)	0.249
Location (right/left lobe of liver)	14(53.8)/12(46.2)	19(73.1)/7(26.9)	0.150
Echogenicity (hypo-/hyper-/mix-echoic)	6(23.1)/17(65.4)/3(11.5)	19(73.1)/5(19.2)/2(7.7)	0.001
Homogeneity (homogeneous/heterogeneous)	8(30.8)/18(69.2)	10(38.5)/16(61.5)	0.560
Shape (regular/lobulated)	20(76.9)/6(23.1)	21(80.8)/5(19.2)	0.734
Margin (well-/ill-defined)	19(73.1)/7(26.9)	22(84.6)/4(15.4)	0.499
Halo sign (yes/no)	5(19.2)/21(80.8)	8(30.8)/18(69.2)	0.337
Blood flow signals (short linear/spot-like/no)	9(34.6)/12(46.2)/5(19.2)	13(50.0)/11(42.3)/2(7.7)	0.358
Resistance index	0.67(0.54, 0.72)	0.61(0.59, 0.70)	0.547
CEUS arterial phase			
Enhancement intensity			1.000
hyperenhancement	25(96.2)	26(100.0)	
isoenhancement	1(3.8)	0	
hypoenhancement	0	0	
Enhancement pattern			0.352
homogeneous	8(30.8)	9(34.6)	
rim	2(7.7)	0	
not rim, not peripheral, discontinuous, heterogeneous	16(61.5)	17(65.4)	
Portal venous phase (-/iso-/hypo-enhancement)			0.312
hyperenhancement	2(7.7)	0	
isoenhancement	9(34.6)	8(30.8)	
hypoenhancement	15(57.7)	18(69.2)	
Late phase (hyper-/iso-/hypo-enhancement)			0.492
hyperenhancement	1(3.8)	0	
isoenhancement	2(7.7)	1(3.8)	
hypoenhancement	23(88.5)	25(96.2)	
Washout time (<60 s/≥60 s)	1(3.8)/25(96.2)	3(11.5)/23(88.5)	0.610
Washout intensity (mild/marked)	22(84.6)/4(15.4)	24(92.3)/2(7.7)	0.668
CEUS LI-RADS category			0.313
LR-3	1(3.8)	0	
LR-4	3(11.5)	2(7.7)	
LR-5	20(76.9)	24(92.3)	
LR-M	2(7.7)	0	

**Table 3 diagnostics-13-01337-t003:** The morphologic and enhancement features of SH-HCCs and non-SH-HCCs as seen via MRI (%).

Features	SH-HCC (*n* = 26)	Non-SH-HCC (*n* = 26)	*p* Value
Signal intensity on T1WI			0.568
hyperintensity	21(80.7)	22(84.6)	
isointensity	3(11.5)	1(3.8)	
hypointensity	0	1(3.8)	
mixintensity	2(7.7)	2(7.7)	
T1WI opposed-phase signal drop			0.000
yes	22(84.6)	7(26.9)	
no	4(15.4)	19(73.1)	
Signal intensity on T2WI			0.972
hyperintensity	25(96.2)	23(88.5)	
isointensity	0	1(3.8)	
hypointensity	0	0	
mixintensity	1(3.8)	2(7.7)	
Signal homogeneity on T2WI			0.337
homogeneous	5(19.2)	8(30.8)	
heterogeneous	21(80.8)	18(69.2)	
Signal in DWI			0.124
hyperintensity	24(92.3)	25(96.2)	
isointensity	2(7.7)	0	
hypointensity	0	1(3.8)	
Arterial enhancement intensity			0.308
hyperenhancement	23(88.5)	24(92.3)	
isoenhancement	1(3.8)	2(7.7)	
hypoenhancement	2(7.7)	0	
Enhancement homogeneity			0.337
homogeneous	5(19.2)	8(30.8)	
heterogeneous	21(80.8)	18(69.2)	
Enhancement pattern			0.283
rim	2(7.7)	1(3.8)	
peripheral discontinuous	2(7.7)	0	
not rim, not peripheral discontinuous	22(84.6)	25(96.2)	
Portal venous enhancement intensity			0.264
Hyper- or iso-enhancement	14(53.8)	9(34.6)	
hypoenhancement	12(46.2)	17(65.4)	
Delayed enhancement intensity			0.465
hyper-or iso- enhancement	6(23.1)	3(11.5)	
hypo-enhancement	20(76.9)	23(88.5)	
Tumor capsule			0.499
no	7(26.9)	4(15.4)	
incomplete or complete	19(73.1)	22(84.6)	
Fat in mass			0.000
diffuse	15(57.7)	0	
focal	11(42.3)	5(19.2)	
no	0	21(80.8)	
Hemorrhage in mass			0.350
yes	1(3.8)	4(15.4)	
no	25(96.2)	22(84.6)	
Necrosis in mass			0.703
yes	3(11.5)	5(19.2)	
no	23(88.5)	21(80.8)	
CT/MRI LI-RADS category			0.680
LR-3	4(15.4)	3(11.5)	
LR-4	7(26.9)	6(23.1)	
LR-5	14(53.8)	17(65.4)	
LR-M	1(3.8)	0	

**Table 4 diagnostics-13-01337-t004:** Histopathological characteristics of SH-HCCs and non-SH-HCCs (%).

Variables	SH-HCC (*n* = 26)	Non-SH-HCC (*n* = 26)	*p* Value
Edmondson grade of HCC			0.245
Grade II	12(46.2)	7(26.9)	
Grade III	14(53.8)	18(69.2)	
Grade IV	0	1(7.7)	
MVI status			0.025
M0	15(57.7)	7(30.8)	
M1 + M2	11(42.3)	19(73.1)	
Ki-67; (/)	/	/	0.095
<20%	15(57.7)	9(34.6)	
≥20%	11(42.3)	17(65.4)	
Satellite lesions			0.490
yes	26(100.0)	24(92.3)	
no	0	2(7.7)	
Vascular invasion			0.610
yes	1(3.8)	3(11.5)	
no	25(96.2)	23(88.5)	
Lymph node metastasis			0.490
yes	0	2(7.7)	
no	26(100.0)	24(92.3)	
Hepatic stestosis			0.000
>5%	23(88.5)	7(26.9)	
≥5%	3(11.5)	19(73.1)	
Liver fibrosis stage			0.044
S0-S2	20(76.9)	13(50.0)	
S3-S4	6(23.1)	13(50.0)	
Liver necroinflammatory activity			0.569
G0-G2	17(65.4)	15(57.7)	
G3-G4	9(34.6)	11(42.3)	

Note: MVI: microvascular invasion.

## Data Availability

The data presented in this study are available on request from the corresponding author. The data are not publicly available due to ethical implications.

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
