# Peer review of "Contrast-Enhanced Imaging Features and Clinicopathological Investigation of Steatohepatitic Hepatocellular Carcinoma"

_diagnostics, 2023, doi:10.3390/diagnostics13071337_

Round 1
Reviewer 1 Report
Authors intent is to explore the CEUS and CEMRI features as well as clincopathological characteristics of SH-HCCC in comparison with non SH-HCC.
Although the sample size is low and some selection bias for sure exist. i belive the data presented is of high quality. Images and figures are clear and good quality.
Statistical analysis is correctly used.
Authors should state in the intruduction a general paragraph on CEUS, in particular the application in other fields.
Please consider the following papers:
Contrast-Enhanced Ultrasound (CEUS) in the Evaluation of Renal Masses with Histopathological Validation—Results from a Prospective Single-Center Study. Diagnostics 2022, 12(5), 1209; https://doi.org/10.3390/diagnostics12051209
The Value of Contrast-Enhanced Ultrasound (CEUS) in Differentiating Testicular Masses: A Systematic Review and Meta-Analysis. Appl. Sci. 2021, 11(19), 8990; https://doi.org/10.3390/app11198990
Author Response
Dear Editor and Reviewers:
On behalf of my co-authours, we are very grateful to your giving us an opportunity to revise our manuscript. We appreciate you very much for your positive and constructive comments and suggestions on our manuscript entitled” Contrast Enhanced Imaging Features and Clinicopathological Investigation of Steatohepatitic Hepatocellular Carcinoma”
Response to the comments of Reviewer #1
Authors intent is to explore the CEUS and CEMRI features as well as clincopathological characteristics of SH-HCCC in comparison with non SH-HCC.
Although the sample size is low and some selection bias for sure exist. i belive the data presented is of high quality. Images and figures are clear and good quality.
Statistical analysis is correctly used.
Response:Thanks for your valuable comment.
Authors should state in the intruduction a general paragraph on CEUS, in particular the application in other fields.
Please consider the following papers:
Contrast-Enhanced Ultrasound (CEUS) in the Evaluation of Renal Masses with Histopathological Validation—Results from a Prospective Single-Center Study. Diagnostics 2022, 12(5), 1209; https://doi.org/10.3390/diagnostics12051209
The Value of Contrast-Enhanced Ultrasound (CEUS) in Differentiating Testicular Masses: A Systematic Review and Meta-Analysis. Appl. Sci. 2021, 11(19), 8990; https://doi.org/10.3390/app11198990
Response:Thanks for your nice question. we have added the we have added the application of CEUS in the Introduction of page 2, line 18-23.

Reviewer 2 Report
This is a single center retrospective study evaluating the clinicopathological features of steatohepatitic hepatocellular carcinoma. I have several comments.
1. In abstract, there are many abbreviations which are not spelled out. Please spell out abbreviations at initial appearance.
2. I don't understand the definition of steatohepatitic HCC. Findings of steatohepatitis in background liver is not the definition?
Author Response
Response to the comments of Reviewer #2
- In abstract, there are many abbreviations which are not spelled out. Please spell out abbreviations at initial appearance.
Response:Thank you very much your nice reminder. We have spelled out abbreviations at initial appearance of abstract.
- I don't understand the definition of steatohepatitic HCC. Findings of steatohepatitis in background liver is not the definition?
Response:Thanks for your valuable comment. Steatohepatitic hepatocellular carcinoma (SH-HCC) is a distinctive histologic variant for the presence of steatohepatitis, including ballooning of malignant hepatocytes, in-flammation, Mallory-Denk bodies, and pericellular fibrosis inside the tumour. It has been reported to be tightly associated with HCV infection, Type 2 diabetes mellitus (T2DM), obesity, and underlying MAFLD. Nonetheless, some researcher identified partial SH-HCCs patients who were absence of fatty liver may be driven by tumor-specific genetic alterations. They commented that SH-HCC can exist independently of steatohepatitic background.

Round 2
Reviewer 2 Report
The authors have revised the manuscript appropriately.